# Cell Death-Related Ubiquitin Modifications in Inflammatory Syndromes: From Mice to Men

**DOI:** 10.3390/biomedicines10061436

**Published:** 2022-06-17

**Authors:** Nieves Peltzer, Alessandro Annibaldi

**Affiliations:** 1Center for Molecular Medicine Cologne (CMMC), University of Cologne, Robert-Koch-Strasse 21, 50931 Köln, Germany; 2Department of Translational Genomics, University of Cologne, Weyertal 115b, 50931 Köln, Germany; 3Cologne Excellence Cluster on Cellular Stress Responses in Aging-Associated Diseases (CECAD) Research Center, University of Cologne, Joseph-Steltzmann-Strasse 26, 50931 Köln, Germany

**Keywords:** cell death, apoptosis, necroptosis, pyroptosis, ubiquitin, LUBAC, OTULIN, A20, inflammation, autoimmunity, human genetics

## Abstract

Aberrant cell death can cause inflammation and inflammation-related diseases. While the link between cell death and inflammation has been widely established in mouse models, evidence supporting a role for cell death in the onset of inflammatory and autoimmune diseases in patients is still missing. In this review, we discuss how the lessons learnt from mouse models can help shed new light on the initiating or contributing events leading to immune-mediated disorders. In addition, we discuss how multiomic approaches can provide new insight on the soluble factors released by dying cells that might contribute to the development of such diseases.

## 1. Introduction

The field of cell death has undergone quite a substantial evolution over the past three decades. During the 1990s and the first years of the new century, scientists were mainly focused on how the knowledge of cell death pathways could have helped improve cancer therapy, to eliminate as many cancer cells as possible, with little interest for the pathophysiological roles of cell death [1,2]. In the last decade or so, the field has moved towards a new direction, with the purpose to understand how the decision between life and death regulates tissue homeostasis and inflammatory responses during tissue damage or pathogen infection [3,4]. It became increasingly clear that cell death-regulating molecules are not always programmed to kill and can fulfil cell death-independent functions [5,6]. In addition, cell death is not a biological end point and, in the process of dying or after their death, cells can still emit signals in a programmed manner [7]. These signals evoke inflammatory programs that are essential for the ability of tissues to recover from different types of insults and restore homeostasis [8]. However, aberrantly regulated cell death can exacerbate inflammatory processes that can in turn cause tissue failure and inflammatory disorders and autoimmunity [9]. Therefore, the magnitude of cell death processes is always fine-tuned by multiple control mechanisms that are in place to prevent the detrimental effects of uncontrolled cell death [10].

Cytokines of the tumour necrosis factor (TNF) family are crucial regulators of cell death, inflammation and autoimmunity [11]. TNF receptor 1 (TNFR1) is a member of the death receptor (DR) family. These receptors are characterised by the presence, in their intracellular portion, of a death domain (DD) that is able to initiate cell death cascades [11]. Other well-studied members of this family are CD95 (Fas/APO-1), TNF-related apoptosis-inducing ligand (TRAIL)-R1 (DR4) and TRAIL-R2 (DR5) [12]. It is the research conducted on the TNF-TNFR1 system during the past two decades that allowed scientists to discover the previously unappreciated link between cell death and inflammation. Indeed, the numerous mouse models developed in the past 10 years, bearing inactivating mutation in players on the TNFR1-signalling pathway, revealed that cell death could be the triggering event in inflammatory and autoimmune diseases [9]. In this review, we focus on different mouse models that led to the discovery of the relationship between cell death and inflammation, how they contributed to establish the link between cell death and inflammation-related disorders and how these disorders resemble human autoinflammatory and autoimmune diseases. In addition, we review the latest omic-based approaches adopted to elucidate the inflammatory potential of dying cells.

## 2. TNFR1-Signalling Pathway

Upon binding to its cognate ligand, TNF, TNFR1 trimerizes and initiates the formation of a receptor-bound complex called complex-I or TNFR1-associated signalling complex (TNFR1-RSC) [13]. The first event for complex-I formation is the recruitment of the adaptor protein TRADD, and the kinase RIPK1, via DD-interaction [14]. Subsequently, TRADD recruits TRAF2 that in turn binds to the E3 ligases cIAP1 and cIAP2 [15]. These cIAPs synthesise ubiquitin chains of different topologies (i.e., K63, K48 and K11) on different components of complex-I, including themselves and RIPK1 [16,17]. These ubiquitin chains serve as a scaffold to recruit another E3 ligase, the linear ubiquitin chain assembly complex, LUBAC [18]. LUBAC is a tripartite E3 ligase complex formed of HOIP, HOIL-1 and SHARPIN [19,20]. LUBAC conjugates linear ubiquitin chains, also called methionine (M) 1 chains, to several complex-I components, including RIPK1 and TNFR1 [21]. The ubiquitin chains formed by cIAP1/2 and LUBAC stabilise complex-I and favour the recruitment of different kinase complexes: TAB2/TAB3/TAK1 [22,23,24], NEMO/IKKα/IKKβ and NEMO/TANK1/NAP1/TBK1/IKKε [25]. The TAK1 and IKKα/IKKβ kinases are required for the activation of NF-κB and MAPKs for the expression of pro-survival and pro-inflammatory genes that are required to mount an innate immune response [26]. TAK1 also controls MK2 activation. The M1 chains synthesised by LUBAC also serve to recruit deubiquitinating enzymes (DUBs), namely A20 and CYLD, this last via the adapter SPATA2, which have opposing effects in complex-I [21,27]. While CYLD hydrolyses ubiquitin chains, prevalently K63 chains [28], to control the extent of NF-κB activation, A20 shields them and prevents their removal, ensuring complex-I stability [21] (Figure 1).

The TNF-signalling pathway is tightly controlled by a number of checkpoints that rely on ubiquitination-, phosphorylation-, gene-expression-dependent and proteolytic events [10,29]. In conditions where any of these checkpoints is disabled, there is the formation of a secondary cytoplasmic complex, referred to as complex-II, which is composed of FADD, Caspase-8, cFLIP, RIPK1 and RIPK3, and that has cytotoxic activity [10,13,29]. Complex II can induce (i) apoptosis, via the activation of the initiator Caspase-8, which, in turn, cleaves and activates the executioner Caspase-3 and Caspase-7 [30]; (ii) necroptosis, mediated by the kinase activity of RIPK1 and RIPK3 and the pseudokinase MLKL, which, following RIPK3-mediated activation, forms pores in the plasma membrane [31,32,33]; and (iii) pyroptosis, following Caspase-8-mediated cleavage of Gasdermin D that, similarly to MLKL, has the ability to form pores on the plasma membrane, leading to a lytic type of death [34,35]. While in non-immune cells Caspase-8 activation mainly leads to apoptosis, in innate immune cells, such as macrophages, Caspase-8 activation can induce both caspase-dependent apoptosis and Gasdermin D-mediated pyroptosis-like death [35] (Figure 1).

Both the conjugation and hydrolysis of ubiquitin moieties represent one of the most studied checkpoint mechanisms that control complex-II formation and restrain TNF cytotoxicity [36]. Amongst the key proteins and protein complexes responsible for these ubiquitin-system-mediated control mechanisms, are LUBAC (E3 ligase complex) [37], XIAP (X-linked IAP) [38], A20 and OTULIN (DUBs) [21] (Figure 2). In the next sections we focus on the pathological consequences of mutations disrupting the activity of the above indicated E3s and DUBs in genetically modified mice and human patients, and highlight the similarities and differences between the two systems.

## 3. LUBAC

Genetic deletion of any of the three LUBAC components causes either absence of linear chains (HOIP and HOIL-1) or reduction in linear chains (SHARPIN) in TNFR1-induced complex-I [39]. The phenotype of mice bearing a naturally occurring mutation in the SHARPIN gene, referred to as *cpdm* (chronic proliferative dermatitis mice) (Table 1), which causes its deletion, initially puzzled scientists of the cell death field [40]. Although SHARPIN deletion caused reduced NF-κB-mediated gene activation in vitro, *cpdm* mice developed chronic dermatitis and multiorgan inflammation [20]. This conundrum was solved by findings showing that (i) attenuation of linear ubiquitination in complex-I on the one side impairs NF-κB activation and reduces gene expression, but on the other side causes cell death by favouring complex-II formation in response to TNF stimulation [20]; and (ii) TNF-mediated cell death can be a potent trigger of inflammation. The latter was supported by the evidence that genetic deletion of TNF rescues *cpdm* mice from developing dermatitis and represented a watershed in the cell death field [20]. Indisputable evidence confirming that cell death was the cause of inflammation in *cpdm* mice came from the fact that combined deletion of FADD in keratinocytes or Caspase-8, to suppress apoptosis, and RIPK3 or MLKL to suppress necroptosis, prevented the inflammatory phenotype of SHARPIN mutant-mice [41,42]. Around the same years, animal models of cell death-induced inflammation, some of which we discuss here, boomed, further corroborating the notion that cell death can be the etiological agent of inflammatory syndromes. Differently from mice bearing the SHARPIN mutation, mice lacking HOIP or HOIL-1 are embryonically lethal due to exacerbated endothelial cell death and heart defects [39,43] (Table 1). Yet, the selective deletion of HOIP or HOIL-1 in keratinocyte causes severe skin inflammation, which is cell death dependent [44,45]. However, differently from *cpdm*, lethal dermatitis was found to be only partially TNF-driven [44]. Indeed, while concomitant deletion of RIPK3 and Caspase-8 completely prevents the inflammatory lesions, loss of TNFR1 delays the onset of dermatitis, but mutant mice still succumb later in life due to severe skin inflammation [44]. What triggers cell death in the absence of linear ubiquitination beyond TNF? It was reported that the inflammatory phenotype occurring as a consequence of HOIL-1 deficiency could be significantly delayed by the simultaneous deletion of TNFR1 and the DD of CD95 and TRAIL-R. This indicates that these three death receptors act in concert to induce cell death in the skin when LUBAC activity is completely abrogated [44]. In the liver, HOIP deletion causes hepatocellular carcinoma that arises from inflammation caused by hepatocyte death [46]. The role of LUBAC in immune cells has also been described. T cell-specific deletion of HOIP or HOIL-1 leads to an almost complete depletion of CD4/CD8+ T cells in mice [47]. HOIP- and HOIL-1-deficient T cells exhibited delayed NF-κB activation upon TCR and TNFR1 stimulation, consistently with a role for LUBAC in the activation of NF-κB. However, enforced NF-κB activation via the overexpression of a constitutively active version of IKKβ (IKKβca), does not restore normal T cell numbers [47]. Equally intriguing was the fact that T cell developmental defects also seem to be independent of cell death activation. Similarly, loss of LUBAC in B cells impairs signalling via the TNFR superfamily member CD40, highlighting an important role of linear ubiquitination in B cell activation [48]. Notably, mice with full body deficiency in HOIP or HOIL-1 displayed severe defects in haematopoietic progenitors, which affected erythropoiesis, and this was independent of cell death [39,49] (Table 1). This evidence would suggest that, differently from keratinocytes or other cell types, LUBAC activity does not primarily inhibit cell death in immune cells, but it rather plays an important role in coordinating different signals required for cell development or differentiation.

Last, although HOIL-1 also bears E3 ligase activity, it is not essential for LUBAC activity as E3 catalytic inactive mice are viable [18,50,51]. Instead, HOIL-1 catalytic activity limits linear ubiquitination. Indeed, mouse embryonic fibroblasts (MEFs) expressing catalytically inactive HOIL-1 are protected from cell death and have enhanced NF-κB activation in response to TNF due to increased levels of linear ubiquitin chains [39,51]. As a consequence, HOIL-1 catalytic inactive mice are protected from hepatocyte death in a model of liver damage and are also protected from dermatitis in a *cpdm* background [51]. Intriguingly, mice harbouring HOIL-1 catalytic inactivity displayed increased glycogen deposition in muscle [52].

To date, only a limited number of patients bearing LUBAC mutations have ever been reported. The amino acid sequence similarity between human and mouse is 91.6% for HOIL-1, 86.5% for HOIP and 73.6% for SHARPIN. The first reported LUBAC-mutant patients, in 2012, were two sisters from a non-consanguineous marriage, with compound heterozygous mutation of HOIL-1, consisting of deletion and nonsense mutation (p.Q185X) and one boy from a consanguineous marriage, with homozygous deletion of two nucleotides (c.121_122delCT) in HOIL-1 [53]. Few weeks after their birth, they started developing a series of disorders including autoinflammation (e.g., abdominal pain), immunodeficiency, which rendered the three patients susceptible to bacterial infections, amylopectin-like deposits in muscle and cardiomyopathy [53]. They all died by the age of eight years [53]. Immune cells, in particular the monocytes, lacking HOIL-1 are hyperresponsive to IL-1β stimulation, with exaggerated cytokine production, including IL-6 and IL-8. On the other hand, non-immune cells, such as fibroblasts, exhibited a delayed NF-κB response following IL-1β and TNF stimulation and severely impaired cytokine and chemokine production [53] (Table 1). This dichotomy most likely explains the paradoxical clinical phenotype observed in the three HOIL-1 mutant patients. Indeed, while the hyperresponsiveness of monocytes is the underlying cause of autoinflammation, the refractoriness of non-immune cells to mount an innate immune response could explain the immunodeficiency and susceptibility to bacterial infections. One year later, two independent studies reported a few more cases of patients with homozygous or heterozygous compound truncating or missense mutations of HOIL-1 [54]. These patients suffered from progressive muscular weakness, abnormal accumulation of glycogen in muscles and cardiomyopathy. Intriguingly, they presented no sign of autoinflammation and immunodeficiency [54]. Even more intriguingly was the discovery of two more patients, in 2018, carrying HOIL-1 mutations with both autoinflammatory/immunodeficiency and myopathic features [55]. At present, it is not known what determines whether HOIL-1 mutant patients preponderantly have one or the other phenotype, or both. It is also difficult to understand whether the mutations present in these patients would result in reduced HOIL-1 E3 ligase activity or whether they would rather behave as a linear ubiquitin-null mutant.

The first HOIP homozygous missense mutation (L72P) patient was reported in 2015 [56]. This patient was born from consanguineous parents. The second one, born from non-consanguineous parents, carrying biallelic variants, was identified in 2019 [57]. In both cases, the authors reported that, similarly to some HOIL-1 mutant patients, fibroblasts have impaired NF-κB activation upon TNF and IL-1β stimulation while monocytes are hyperresponsive to IL-1β stimulation [56]. In both cases, patients presented clinical features characteristic of multiorgan autoinflammation and immunodeficiency (recurrent bacterial infection). Lymphopenia (T cell depletion) was only present in the first observed patient [56]. The similar clinical manifestations between HOIL-1 and HOIP mutant patients can probably be attributed to the loss of linear ubiquitination, which is a common feature of loss of these two proteins, at least in mice [39] (Table 1).

Very recently, a non-synonymous variant of SHARPIN was identified as genetic risk factor for LOAD (late-onset Alzheimer’s disease) in a cohort of 202 Japanese individuals [58]. This variant has an amino acid substitution (G186R) that seemingly affects its subcellular localisation and NF-κB activation. In a follow-up study, six more SHARPIN variants were identified from a cohort of 180 patients with LOAD and 184 patients with mild cognitive impairment (Table 1). This, at present, is the only reported association between SHARPIN mutation and human diseases [59].

The different LUBAC mutant mouse models described above have been instrumental in unveiling the physiological role of linear ubiquitination and how linear ubiquitin chains orchestrate inflammatory programs. Equally important was the fact that they allowed to unravel the link between cell death and inflammation and the potent inflammatory potential of cell death activation. Mouse work revealed that the individual LUBAC components contribute to optimal gene activation following stimulation of immune receptors, including TNFR1, but, most importantly, they limit the killing activity of TNF. Therefore, the predominant phenotypic effect triggered in mice by their absence is cell death and cell death-induced inflammation. However, the phenotypic similarities between LUBAC mutant mice and patients are limited to some features of glycogen deposition and heart defects. In the human setting, loss of function mutations of HOIP, HOIL-1 or SHARPIN do not only result in autoinflammation, but also in immunodeficiency, glycogen storage disorders (HOIP and HOIL-1) and neurodegeneration (SHARPIN). This might indicate that (i) in humans, the gene activatory functions of LUBAC are as or more predominant than its cell death inhibitory functions, although the occurrence of cell death has not been fully analysed in patients; and (ii) those individuals carrying LUBAC mutations that escape lethality in utero, might have backup systems in place to regulate cell death and inflammation that are not completely dependent on LUBAC. An extensive cell death analysis in LUBAC mutant patients, using classical cell death markers might help elucidate the role of LUBAC in controlling organism homeostasis, and find therapeutic strategies to improve the care of LUBAC mutant patients.

## 4. OTULIN

OTULIN is the only known linear chain-specific deubiquitinase [81]. The amino acid sequence similarity between human and mouse OTULIN is 90.1%. In 2016, homozygous missense mutation of OTULIN was identified in three siblings, from a consanguineous family, affected by a severe sterile form of autoinflammation, which was named OTULIN-related autoinflammatoy syndrome (ORAS) [61]. In the same year, another group identified three more patients, from three different consanguineous families, carrying OTULIN biallelic mutations and symptoms of systemic sterile inflammation (e.g., prolonged fevers and diarrhoea) that they called Otulipenia [63] (Table 1). Few more patients carrying compound heterozygous mutations on OTULIN were identified, with similar clinical manifestations [62]. This prompted different groups of scientists to investigate the molecular basis of this mutant OTULIN-driven inflammatory disorder using mouse models. The most intuitive explanation as to why OTULIN-mutant patients suffered from a severe inflammatory disease was that, without functional OTULIN, there would be an excess of linear chains that would in turn exacerbate NF-κB responses and the consequent production of pro-inflammatory factors. Indeed, both OTULIN-mutant patients and OTULIN-deficient mice (conditional full body and myeloid cell-specific) exhibited linear chains accumulation, increased NF-κB activation and excessive cytokine production [61,63]. The fact that Infliximab (TNF-neutralising antibody) drastically reduced the inflammatory syndrome in patients and mice [61] indicated that the main instigator of the excessive NF-κB activation and cytokine production observed in absence of functional OTULIN (patients) and full-length protein (mice) is TNF (Table 1). Intriguingly, it was concluded that, differently from LUBAC deficient mice, it was the gene activation ability of TNF rather than its cell death-promoting potential that caused the inflammatory phenotype in mice lacking OTULIN. 

This view was subsequently challenged by another study that, using a catalytically inactive mutant of OTULIN in mice, showed that the primary effect of linear chains accumulation is not NF-κB hyperactivation but rather complex-II-mediated cell death, in the form of apoptosis and necroptosis [64] (Table 1), the reason for this being that absence of OTULIN causes hyperubiquitination of the LUBAC components that impairs their recruitment to complex-I (Figure 1) [21,64]. This would in turn favour complex-II formation and cell death. This scenario was later confirmed by reports showing that OTULIN deletion in the liver causes TNFR1-driven, apoptosis- and compensatory proliferation-mediated liver pathology, while OTULIN deletion in keratinocytes causes TNFR1-driven, RIPK1 kinase activity-mediated, cell death-dependent skin inflammation [65,66] (Table 1). Of note, some OTULIN-mutant patients display signs of liver dysfunction and skin inflammation in the form of panniculitis and neutrophilic dermatosis [67]. Despite the contrasting results concerning the etiological agent of the systemic inflammatory syndrome that characterises OTULIN mutant mice (NF-κB hyperactivation vs. cell death), the common denominator of these mouse models is that, similarly to human settings, the blockade of the TNF/TNFR1 system significantly ameliorates the disease.

Importantly, OTULIN has also been implicated in signalling events that are different from cell death and inflammation per se. For example, the *Gumby* mutation, which is a spontaneous mutation in OTULIN (W96R), results in embryonic lethality resembling the report on OTULIN catalytically inactive mice [60]. Gumby mice display increased Wnt signalling [60]. Whether this is the cause for lethality remains unresolved. In addition, the pathology of OTULIN deficiency in the liver seems to be independent of TNFR1 signalling but dependent on aberrant mTOR activation [67].

The OTULIN-mutant mouse models were extremely useful to unveil the link between OTULIN mutations and excessive linear chains in ORAS/Otulipenia. In addition, they allowed to understand why infliximab has such therapeutic benefits in patients. At present, and similar to LUBAC-mutant patients, evidence that OTULIN absence can unleash inflammatory cell death is still missing. One could speculate that absence of OTULIN activity can induce cell death only in some cell types, while in others sustained NF-κB activation would be the main outcome. Future analysis employing cell death-specific stainings will be required to understand the mechanisms of hyperinflammation and the respective contributions of cell death and NF-κB in OTULIN mutant patients.

## 5. A20

A20 is a deubiquitinase enzyme that exhibits 88.1% amino acid sequence conservation between human and mouse. It was identified in 1990 as a NF-κB target gene [82], which had the ability of preventing TNF-induced cytotoxicity [83]. It was subsequently discovered that A20 is not only a NF-κB target gene but also an inhibitor of the NF-κB-signalling pathway [84]. Consistent with this idea, A20 null mice die perinatally due to multiorgan inflammation [84]. Intriguingly, the ability of A20 to suppress inflammation does not reside in its deubiquitinase activity, since mice carrying a point mutation in A20 catalytic domain do not display inflammation [85]. After the realisation that A20 deubiquitinase activity is dispensable to control inflammation, different groups have tried to uncover which other domains of A20 are responsible for this. A20 is a ubiquitin-editing enzyme, which not only possesses DUB activity but also E3 ligase activity, mediated by the fourth zinc finger domain (ZnF4) [68]. Surprisingly, the E3 ligase activity is not required to keep inflammation in check, since ZnF4 mutant mice are viable and healthy [86,87]. A major advance in the understanding of the role of A20 in repressing inflammatory processes came from studies whereby the ZnF7 was mutated. It was shown that the ZnF7 is required for the ability of A20 to suppress NF-κB activation, in a non-catalytic fashion [88]. A subsequent work proved that A20 stabilises linear chains by direct ZnF7-mediated binding; indeed, in the absence of A20, complete absence of linear chains was observed in complex-I [21,89]. Therefore, a model was proposed whereby A20, via the ZnF7 domain, binds to and shields linear chains, preventing their hydrolysis by other DUBs (e.g., CYLD), ensuring complex-I stability. At the same time, this shielding prevents the excessive recruitment of NF-κB-activating molecules to complex-I, such as the NEMO/IKK complex, thereby controlling NF-κB activation [21,90]. Consistently with this model, ZnF7 mutant mice display spontaneous inflammation [76]. Similar to the various LUBAC-mutant mice, the *Tnfaip3^−/−^*, the *Tnfaip3^myel-KO^* (A20 full knock-out or selectively in myeloid cells, respectively) and *Tnfaip3^ZnF7mut^* mutant mice were extremely useful to strengthen the link between cell death and inflammation, and to gain a better understanding of the cause of the inflammatory diseases displayed by A20 mutant patients. A20-deficient mice suffered from severe multiorgan inflammation, including, but not limited to, liver, kidneys and joints. Similarly, although milder, mice lacking A20 in myeloid cells and those bearing a ZnF7 mutant version of A20 developed arthritis [76] (Table 1). Importantly, it has been proven at the genetic level that the cause of arthritis in these mice is not hyperactivation of NF-κB, but rather RIPK1/RIPK3/MLKL-dependent necroptosis of macrophages. This leads in turn to NLRP3 inflammasome activation within the same dying macrophages with the consequent release of IL-1β. Excessive production of IL-1β then causes cartilage erosion and joint inflammation [76,91].

In humans, it has been known for two decades that A20 is a susceptibility gene for autoinflammatory diseases such as systemic lupus erythematosus (SLE), rheumatoid arthritis, psoriasis and diabetes [70,71,72,73,74,75]. However, it was only in 2016 that it was proven that loss-of-function germline mutations in A20 cause systemic autoinflammatory disease [69]. The authors of this study identified five heterozygous truncating mutations in five families. Patients carrying the mutations displayed a range of clinical manifestations including early-onset systemic inflammation, arthritis, oral and genital ulcers, SLE-like disease and central nervous system vasculitis. This systemic inflammatory disease caused by A20 haploinsufficiency was named HA20. Patient-derived immune cells had a strong inflammatory signature (e.g., elevated levels of TNF, IL-6 and IL-17) and were hyperresponsive to inflammasome activation following LPS stimulation. Along the same line, patient-derived fibroblasts exhibited increased NF-κB activation upon TNFR1 stimulation [69] (Table 1).

A20 is another example of how mouse models can be extremely valuable to accelerate the understanding of (i) the genetic cause of a human pathology, or group of pathologies, and (ii) the molecular mechanisms driving the pathology. At the same time, the genetic studies in humans can indicate how to refine the existing mouse models to develop better preclinical disease models. For example, although the A20 mutant mice develop a set of diseases that quite closely recapitulate the patient’s clinical features, it is surprising that almost all the human mutations are found in the OTU catalytic domain (catalytic inactive A20 mutant mice are normal), and no mutation has ever been found in the ZnF7 domain [92]. One explanation could be that in patients there is very little to no A20 detected in lysates from fibroblasts or PBMCs, suggesting that the mutations in the OTU destabilise A20 rather than solely killing its catalytic activity [69]. This possibility could be addressed by generating new genetically modified mice bearing the human corresponding A20 mutations. In addition, the fact that in mice the individual mutations in the OTU and ZnF4 do not trigger the inflammatory phenotype [86,87] could suggest that the combination of the two mutations might induce the phenotype observed in *Tnfaip3^−/−^* and *Tnfaip3^ZnF7mut^* mice. These new potential mouse models would broaden the possibility to study HA20 and find novel therapeutic approaches.

Finally, while the mouse models clearly indicated that the aetiology of the systemic inflammation observed in A20 mutant mice is cell death, in patients there is still lack of evidence supporting this possibility. Similar to what was highlighted for LUBAC and OTULIN mutant patients, cell death marker stainings, (phospho)proteomic and ubiquitinome analysis on patients’ samples might help determine the contribution of cell death vs. NF-κB hyperactivation to the disease.

## 6. XIAP

XIAP is an E3 ligase enzyme that belongs to the IAP (inhibitor of apoptosis) family [93]. Human and mouse XIAP share 89.3% of their amino acid sequence. It was initially characterised as able to inhibit Caspase-9, by preventing its dimerization, and Caspase-3 and Caspase-7, by blocking their active site [94,95,96,97]. However, it subsequently became clear that XIAP is an important immune regulator, both in a cell death-independent and -dependent manner. Indeed, its E3 ligase activity plays a crucial role in pathogen responses mediated by NOD2, a member of the NLR family [98]. In the NOD2-signalling pathway, XIAP-mediated ubiquitination of RIPK2 is crucial for the correct activation of the pathway and secretion of the cytokines needed for the pathogen response [98]. Therefore, the importance of XIAP in the NOD2 pathway is independent of its ability to regulate cell death. By contrast, following TLR activation (e.g., TLR2 or TLR4), XIAP is required to prevent, in myeloid cells, RIPK3-mediated necroptosis and the concomitant NLRP3 activation and IL-1β release [99] (Figure 2 and Table 1). Additionally, although XIAP is not recruited to TNF-induced complex-I, it regulates RIPK1 ubiquitination outside complex-I, therefore contributing to the regulation of complex-II formation [38] (Figure 1). In 2006, mutations in XIAP were found in 12 individuals belonging to three different families affected by X-linked lymphoproliferative syndrome (XLP) [79]. This was the first time that XIAP mutations were associated to a human disease. To date, many XIAP mutations have been identified, recently summarised in [100], which include nonsense and missense mutations, exon deletion, and small insertions and deletions, often leading to premature stop codon and protein deficiency. XLP is a rare immunodeficiency that is characterised by hemophagocytic lymphohystiocystosis (HLH), hypogammaglobulinaemia and lymphoma [101]. This syndrome normally develops following Epstein–Barr virus infection. The identification of more XIAP-mutant patients in the following years prompted clinicians to classify XIAP deficiency-caused disease as familial HLH (FHHL) or XLP2 [102] (Table 1). XLP2 differs from XLP1 in some immunological features, including absence of lymphoma development and high risk of IBD (inflammatory bowel disease). In particular, IBD is observed in about 25% of XIAP mutant patients and it is often refractory to treatment and lethal in 10% of the cases. Moreover, XIAP mutations are detected in up to 4% of male paediatric patients with very early onset IBD [80,103]. Another immunologic feature of XLP2 patients is the excessive activation of macrophages and dendritic cells to EBV and other viruses [101]. However, the mechanisms linking XIAP mutations to XLP2 disease and intestinal epithelial barrier damage are not yet entirely understood. Given the fact that XIAP plays a crucial role in the NOD2 pathway and NOD2 was the first ever identified risk gene for IBD, one could argue that XIAP deletion would predispose patients to IBD by impairing the NOD2 pathway. This speculation would be supported by the evidence that the majority of the missense mutations on XIAP map either in the BIR2 or the RING domain, both crucial for correct activation of the NOD2-signalling pathway [100]. However, while the penetrance of IBD in XIAP-mutant patients is 23%, only 1.5% of individuals carrying homozygous NOD2 risk variants develop IBD [78]. This suggests that the role of XIAP in ensuring intestinal homeostasis goes beyond its function in the NOD2 pathway, perhaps to its cell death regulatory functions.

Mouse models were again of great help to gain insights into the role of XIAP in the regulation of immune system responses (Table 1). Indeed, two recent works have shed new light on the role of XIAP deficiency in IBD pathogenesis, using *Xiap^−/−^* mice as a model [77,78]. In one case, the authors showed that XIAP-deficient mice have a reduced number of Paneth cells, as consequence of their death, which is TNF- and microbiota-dependent and RIPK1/RIPK3-mediated. Decrease in Paneth cells correlates with a decrease in production and secretion of antimicrobial peptides and change in the structure of the microbiota, termed dysbiosis. These changes, per se, are not enough to elicit intestinal inflammation in mice. However, *Xiap^−/−^* mice are very sensitive to intestinal inflammation triggers, such as DSS or the pathogenic bacterial strain *Helicobacter hepaticus*. Importantly, delivery of antimicrobial peptide to the intestine, by means of adenoviruses, allowed XIAP-deficient mice to clear the *H. hepaticus* [78]. These findings are in line with the fact that in XIAP-mutant patients, similar to the mouse, there is a reduction in Paneth cells in the ileum and often the intestinal inflammation is triggered by bacterial or viral infection. In the other study, the authors have demonstrated that, unlike the abovementioned work, XIAP-deficient mice develop spontaneous ileal inflammation, which is microbiome and TNF dependent. In addition, the authors showed that both TNFR1 and TNFR2 contributed to the inflammation since their individual genetic ablation abrogates the inflammation. Furthermore, they proved that the death of dendritic cells mediated by the TLR/TNFR2/RIPK3 axis ignites the intestinal inflammation in XIAP-null mice [78]. Very interestingly, elevated levels of TNFR2 correlates with disease severity in paediatric patients affected by IBD. These studies are attractive representative examples of how mouse models can help take human genetic studies forward to understand the molecular mechanisms underlying pathological clinical cases. For example, they corroborate the central role of Paneth cell damage in IBD and the importance of TNF as the instigator of the disease. They also hint at the importance of cell death, of both Paneth cells and dendritic cells, in the initiation of the intestinal inflammation in XIAP-mutant patients. Equally important, these mouse-based studies can help devise new therapeutic intervention, such as cell death inhibition, or delivery to the intestine of specific antimicrobial compounds, which will ultimately help improve the patient’s care.

## 7. Omics

Genomic sequencing approaches have decisively contributed to identify the cause of many inflammation-related genetic diseases. The combination of genetic studies with biochemical studies has then helped dissect the molecular mechanisms underlying these genetic diseases and provided sound scientific basis for therapeutic intervention. The question that can be raised now is: How can different omic approaches further help understand the aetiology of human inflammatory diseases? It is established that TNF often plays an eminent role in inflammatory syndromes and that TNF-induced cell death, rather than TNF-induced gene activation, might be the decisive factor for the development of these syndromes. Therefore, the next question to be addressed now relates to the nature of factors released by dying cells that can instigate inflammatory processes.

Different cell death forms have different inflammatory potential. It is widely accepted that, while necroptosis and pyroptosis are inflammatory cell death forms, because of membrane rupture and intracellular content spillage, apoptosis is immunologically silent [3]. Interestingly, it has been demonstrated that necroptotic cells do not only passively release soluble factors as a consequence of plasma membrane rupture, but they are transcriptionally and translationally active for the production of pro-inflammatory cytokines [7]. Conversely, it was reported that apoptotic cells shut down translation via caspases, hence their scarce inflammatory potential [104]. However, some mouse models seem to challenge, at least partially, the current dogma. For example, the skin inflammatory phenotype observed in SHARPIN mutant mice has, at present, to be solely accounted to apoptosis [41,42]. In order to ascertain whether the different inflammatory potential of apoptosis and necroptosis is to be accounted to differences in the soluble factors that are released by the dying cells, Tanzer and colleagues took an unbiased, mass-spectrometry-based approach [105]. With this approach they analysed supernatant of human lymphoma cells and primary human macrophages undergoing TNF-induced apoptosis or necroptosis. As expected, a large number of proteins were significantly associated with either cell death type. Surprisingly, there was no significant qualitative or quantitative difference in terms of conventional cytokines between apoptosis and necroptosis. However, intriguingly, the authors observed that the supernatant of apoptotic cells had high levels of nucleosome components while the supernatant of necroptotic cells had high levels of lysosomal proteins [105]. How this translates into the different inflammatory potential between apoptosis and necroptosis is unknown. Furthermore, it is conceivable to think that only a limited number of factors, differentially released from necroptotic cells with respect to apoptosis, have the potential to trigger inflammation. More refined proteomic analysis to be conducted on more clinically related settings will be needed to try to determine the inflammatory potential of factors specifically secreted by dying cells. Proteomic analysis could be coupled to transcriptomic-based and mass-cytometry-based approaches with the purpose to examine cell death-specific signatures. Reliable animal models that recapitulate human mutation-driven, cell death-dependent diseases would be again the starting point for this combinatorial approach.

## 8. Conclusions

The last decade saw an impressive body of work that enabled us to understand how cell death modulates inflammatory diseases and how this was associated with human genetics. It is not always expected that mouse models completely recapitulate the human settings. However, model organisms spanning from 2D cells/organoids to invertebrates and mice bring us closer to the identification of aetiological factors of chronic inflammation and autoimmune disorders in humans. Excitingly, the gap between mice and men is becoming smaller with the advancement in technologies and preclinical animal models. Currently, mouse work has become the springboard for human studies with the ultimate purpose to design novel therapies that improve the care of patients affected by inflammatory/autoimmune diseases.

## Figures and Tables

**Figure 1 biomedicines-10-01436-f001:**
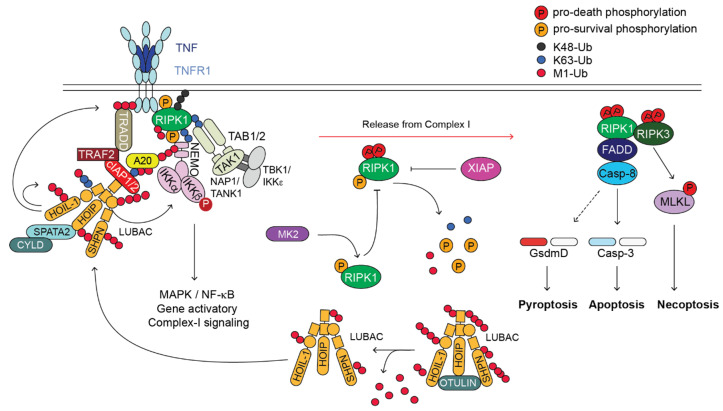
TNFR1-induced-signalling pathway. Cartoon depicting the TNFR1-induced-signalling pathway. Upon binding of TNF to TNFR1, a membrane-bound complex referred to as complex-I forms. This complex is characterised by the presence of adaptor proteins (e.g., TRADD, TRAF2, SPATA2, TAB1/2 and NAP1/TANK1), E3 ligases (e.g., cIAP1/2 and LUBAC), which conjugate poly-ubiquitin chains of different topology (i.e., K63, K48, K11 and M1) to different proteins of the complex, the deubiquitinases (DUBs) A20 and CYLD, and protein kinases such as RIPK1, IKK1/2, TAK1 and TBK1/IKKε. Complex-I promotes the activation of NF-κB and MAPKs that in turn mediate the expression of pro-survival as well as pro-inflammatory genes. Under certain circumstances, a secondary cytoplasmic complex originates in the cytosol from complex-I, referred to as complex-II. This complex is composed of FADD, cFLIP, Caspase-8, RIPK1 and RIPK3. Caspase-8 can trigger apoptosis via activation of Caspase-3 or, in some cell type, pyroptosis, via cleavage of Gasdermin D. Upon Caspase-8 inhibition by the means of synthetic- or viral-encoded caspase inhibitor, RIPK1 activates RIPK3 that in turn phosphorylates MLKL that undergoes activation and executes necroptosis. Of note, OTULIN, XIAP and MK2, have important regulatory functions in the TNFR1-signalling pathway, despite they are not directly recruited to complex-I or complex-II. OTULIN regulates the availability of the LUBAC components for their recruitment to complex-I. XIAP controls RIPK1 ubiquitination status outside complex-II and potentially its cytotoxic activity. MK2, by phosphorylating RIPK1, modulates its killing activity.

**Figure 2 biomedicines-10-01436-f002:**
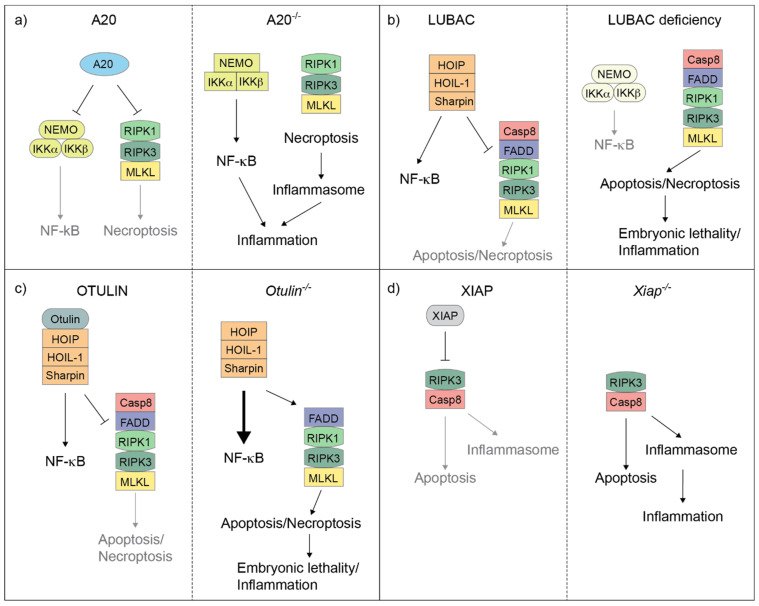
A20, LUBAC, OTULIN and XIAP regulate the balance between cell death and inflammation in mice. Cartoons illustrating how A20 (**a**), LUBAC (**b**), OTULIN (**c**) and XIAP (**d**) control the balance between NF-κB-mediated gene activation and complex-II-mediated cell death in mouse cells. A20 deletion results in the deregulation of both NF-κB response and RIPK1/RIPK3/MLKL-induced necroptosis, which, in turn, triggers inflammasome activation (**a**). Individual deletion of the LUBAC components causes an attenuation of NF-κB response, but an exacerbation of complex-II-mediated cell death, which can result in embryonic lethality or cell death-dependent inflammation in adult mice (**b**). OTULIN deletion leads to hyperactivation of NF-κB and, at the same time, can unleash complex-II-mediated cell death. This can in turn cause embryonic lethality or cell death-dependent inflammation in adult mice (**c**). XIAP deletion causes both Caspase-8-dependent apoptosis and RIPK3-dependent inflammasome activation, which eventually triggers inflammation (**d**).

**Table 1 biomedicines-10-01436-t001:** Overview of the pathological consequences of the deletion or mutation of the indicated genes in mice and human patients.

	Mouse		Human	
Gene	Tissue	Phenotype	Mutation	Phenotype
*Hoip/* *Hoil-1*	Full body deletion	-Embryonic lethality -Excessive endothelial cell death [39,43]	Deletion/ nonsense/ missense	-Autoinflammation (e.g., abdominal pain) -Immunodeficiency (recurrent bacterial infections) -Amylopectin-like deposit in the muscle -Cardiomyopathy -Progressive muscular weakness -Immune cells hyperresponsive to IL-1β stimulation -Delayed NF-kB activation in non-immune cells -Lymphopenia [53,54,55,56]
	Skin-specific deletion	-Excessive keratinocyte death -Severe skin inflammation [44,45]		
	T cell-specific deletion	-Severe T cell depletion -Delayed NF-κB activation in TNF- and TCR-induced-signalling pathways [47]		
*Hoip*	Liver-specific deletion	-Hepatocyte death-driven inflammation -Hepatocellular carcinoma [46]		
	B cell-specific deletion	-Impaired CD40 signalling and antibody production [48]		
*Hoil-1*	Catalytic inactive (Full body)	-Increased NF-κB activation -Protection from cell death -Glycogen deposition [51]		
*Sharpin*	Full body mutation	-Chronic proliferative dermatitis -Liver inflammation -Peyer’s patches loss -Splenomegaly [40]	Missense	Risk factor for LOAD (Late-onset Alzheimer disease) [58,59]
*Otulin*	Full body deletion	-Embryonic lethality -Loss of vascularization [60]	Loss of function	-ORAS (OTULIN-related autoinflammatory syndrome) or Otulipenia -Systemic sterile inflammation (e.g., joint swelling, prolonged fever, diarrhoea, panniculitis) -Developmental delay -Increased linear ubiquitination and NF-κB activation -Infliximab (monoclonal anti-TNF antibody) ameliorates the symptoms [61,62,63]
	Full body conditional deletion	Decreased survival [61]		
	Myeloid-cell specific deletion	-Severe acute systemic inflammation -NF-κB hyperactivation -Excessive cytokine production [61]		
	Catalytic inactivation (full body)	-Embryonic lethality -Deregulated endothelial cell death [64]		
	Skin-specific deletion	-Deregulated keratinocyte death -Severe skin inflammation [65,66]		
	Liver-specific deletion	-TNFR1-driven hepatocyte death -Compensatory proliferation -Hepatocellular carcinoma [67]		
*A20*	Full body deletion	-Perinatal lethality [68] -Cell death-dependent multiorgan inflammation (e.g., liver, kidneys, joints)	Nonsense	-Early onset systemic inflammation (e.g., arthritis, oral and genital ulcers, SLE-like disease, central nervous system vasculitis) -Patients’ derived immune cells have elevated cytokine levels, are hyperresponsive to inflammasome activation. Fibroblasts have increased NF-kB activation following TNF stimulation [69,70,71,72,73,74,75]
	Myeloid cell-specific deletion	-Cell death-dependent joint inflammation [76]		
	ZnF7 mutation (full body)	-Cell death-dependent joint inflammation [76]		
*Xiap*	Full body deletion	-Low grade ileal inflammation, TNFR1- and TNFR2-dependent -Increased sensitivity to pathogenic bacterial strain -Cell death-dependent reduction in Paneth cells and dendritic cells -Cell death is the trigger of ileal inflammation [77,78]	Deletion/ insertion/ nonsense/ missense/ frameshift/ intronic	-Familia hemophagocytic lymphohystiocystiosis (FHLH) or XLP2 -High risk IBD -Excessive activation of macrophages and dendritic cells following EBV infection [79,80]

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
