# Peer review of "Cell Death-Related Ubiquitin Modifications in Inflammatory Syndromes: From Mice to Men"

_biomedicines, 2022, doi:10.3390/biomedicines10061436_

Round 1

Reviewer 1 Report

This is an interesting and well-written review.  Two useful and well-presented figures and a summary table support the text.  The review will provide a useful point of reference for scientists working in this field.  I am happy to recommend publication with just a few minor checks to the current version.  Firstly, there are a number of typographical, grammatical and formatting errors that need to be rectified.  Secondly, the authors refer to TNF throughout the manuscript when, on many occasions, I think they are discussing signalling downstream of TNF-alpha binding to its receptor, and likewise, anti-TNF-alpha therapy, but the Greek letter denoting alpha is missing from the text.  Is there a reason for this?

Author Response

Reviewer #1

This is an interesting and well-written review.  Two useful and well-presented figures and a summary table support the text.  The review will provide a useful point of reference for scientists working in this field.  I am happy to recommend publication with just a few minor checks to the current version.  Firstly, there are a number of typographical, grammatical and formatting errors that need to be rectified.  Secondly, the authors refer to TNF throughout the manuscript when, on many occasions, I think they are discussing signalling downstream of TNF-alpha binding to its receptor, and likewise, anti-TNF-alpha therapy, but the Greek letter denoting alpha is missing from the text.  Is there a reason for this?

We thank this reviewer for the positive comments about our review. We have now revised the manuscript for typographical, grammatical and formatting mistakes.

Concerning the second point: after TNF-b, a cytokine member of the TNF superfamily with the ability to activate TNFR1, was renamed to lymphotoxin-⍺ (LT-⍺), there is no longer the need to denote TNF as TNF-a. Therefore, for simplicity and to be consistent with our published articles, we prefer to use TNF.

Reviewer 2 Report

The review is well structured and summarizes the role of the E3 ligases and DUBs among other interacting proteins. It is an interesting comparison of the studies with mice models and recorded human cases. Below are some comments that could help this paper.

1. It might provide additional insight to the reader if there was some information about the sequence similarity/identity of the proteins mentioned between humans and mice. For example, LUBAC since it was hypothesized to have wider spectrum functions in humans.

2. TNF and TNF are not labeled in Figure 1.

3. Figure 2 needs to be edited. The figure labels and the legend description do not match, and it is difficult to follow. 

4. There seems to a reference missing for the text in page 6 and line 27.

5. Language editing will help with the readability. Mentioned below are a few corrections that should be made among others. 

Page 9, Line 40 - Zing finger domain (ZnF4) – This should be Zinc finger domain.

Page 10, Line 2 - Severe is spelt as sever

Author Response

Reviewer #2

The review is well structured and summarizes the role of the E3 ligases and DUBs among other interacting proteins. It is an interesting comparison of the studies with mice models and recorded human cases. Below are some comments that could help this paper.

We thank this reviewer for the positive comments about our review.

  1. It might provide additional insight to the reader if there was some information about the sequence similarity/identity of the proteins mentioned between humans and mice. For example, LUBAC since it was hypothesized to have wider spectrum functions in humans.

This is a very important comment and we added new sentences describing the sequence homology between mice and men for the three LUBAC components, as well as for OTULIN, A20 and XIAP.

We would like to mention that we removed the phrase in which we state that LUBAC has “wider spectrum functions in humans”. We realized this is very speculative and we decided to discuss hypothesis based on the evidence.

  1. TNF and TNF are not labeled in Figure 1.

Thank you for noting this. We have corrected it accordingly.

  1. Figure 2 needs to be edited. The figure labels and the legend description do not match, and it is difficult to follow. 

We apologize for this mistake. We have corrected the figure and slightly edited the text to make it more reader friendly.

  1. There seems to a reference missing for the text in page 6 and line 27.

We have now added the missing reference.

  1. Language editing will help with the readability. Mentioned below are a few corrections that should be made among others. 

Page 9, Line 40 - Zing finger domain (ZnF4) – This should be Zinc finger domain.

Page 10, Line 2 - Severe is spelt as sever

We have now corrected these typos and revised the manuscript for typographical, grammatical and formatting mistakes.

Reviewer 3 Report

The manuscript by Peltzer&Annibaldi titled ‘Cell death-related Ubiquitin modifications in inflammatory syndromes: from mice to men’ is a well-written, comprehensive review on the tissue necrosis factor-mediated signaling cascade, inflammation, and cell fate with parallels drawn between scientific findings in mouse models and human patients. The manuscript raised no major concerns, except a couple of notes that follow below.

1.      Some molecular players depicted in Figure 1 are not annotated in the legend (i.e., TAB1/2, NAP1, SPATA2). Moreover, NAP1 and SPATA2 are not mentioned in the main text at all. Otulin is depicted as a regulator of LUBAC recruitment to Complex-I but the legend reads ‘recruitment to Complex-II’. Please double check whether any changes are needed to either Figure or the legend.

2.      Some typos are present, e.g. page 5, lane 6, rescue -> rescued?; page 5, lane 34, lIEkwise; page 5, lane 35, ubiquiTNation, to mention a few.

Author Response

Reviewer #3

The manuscript by Peltzer&Annibaldi titled ‘Cell death-related Ubiquitin modifications in inflammatory syndromes: from mice to men’ is a well-written, comprehensive review on the tissue necrosis factor-mediated signaling cascade, inflammation, and cell fate with parallels drawn between scientific findings in mouse models and human patients. The manuscript raised no major concerns, except a couple of notes that follow below.

We thank this reviewer for the positive comments to our review article.

  1. Some molecular players depicted in Figure 1 are not annotated in the legend (i.e., TAB1/2, NAP1, SPATA2). Moreover, NAP1 and SPATA2 are not mentioned in the main text at all. Otulin is depicted as a regulator of LUBAC recruitment to Complex-I but the legend reads ‘recruitment to Complex-II’. Please double check whether any changes are needed to either Figure or the legend.

We thank the reviewer for this spot-on comment, we have now added TAB1/2, NAP1 and SPATA2 in the figure legend and NAP1 and SPATA2 in the main text. We have also corrected the figure legend with respect to OTULIN and its role on LUBAC recruitment to complex-I.

  1. Some typos are present, e.g. page 5, lane 6, rescue -> rescued?; page 5, lane 34, lIEkwise; page 5, lane 35, ubiquiTNation, to mention a few.

We have now corrected these typos and revised the manuscript for typographical, grammatical and formatting mistakes.